# Novel Motion Sequences in Plant-Inspired Robotics: Combining Inspirations from Snap-Trapping in Two Plant Species into an Artificial Venus Flytrap Demonstrator

**DOI:** 10.3390/biomimetics7030099

**Published:** 2022-07-22

**Authors:** Falk J. Tauber, Philipp Auth, Joscha Teichmann, Frank D. Scherag, Thomas Speck

**Affiliations:** 1Plant Biomechanics Group, Botanic Garden, University of Freiburg, 79110 Freiburg, Germany; ph.auth@gmx.de (P.A.); joscha.teichmann@studmail.w-hs.de (J.T.); thomas.speck@biologie.uni-freiburg.de (T.S.); 2Cluster of Excellence livMatS @ FIT-Freiburg Center for Interactive Materials and Bioinspired Technologies, University of Freiburg, 79110 Freiburg, Germany; scherag@imtek.de; 3Laboratory for Chemistry and Physics of Interfaces CPI, Department of Microsystems Engineering-IMTEK, University of Freiburg, 79110 Freiburg, Germany; 4Freiburg Materials Research Center (FMF), University of Freiburg, 79110 Freiburg, Germany

**Keywords:** plant-inspired robotics, artificial Venus flytrap, motion sequence, biomimetics, bioinspiration

## Abstract

The field of plant-inspired robotics is based on principles underlying the movements and attachment and adaptability strategies of plants, which together with their materials systems serve as concept generators. The transference of the functions and underlying structural principles of plants thus enables the development of novel life-like technical materials systems. For example, principles involved in the hinge-less movements of carnivorous snap-trap plants and climbing plants can be used in technical applications. A combination of the snap-trap motion of two plant species (*Aldrovanda vesiculosa* and *Dionaea muscipula*) has led to the creation of a novel motion sequence for plant-inspired robotics in an artificial Venus flytrap system, the Venus Flyflap. The novel motion pattern of Venus Flyflap lobes has been characterized by using four state-of-the-art actuation systems. A kinematic analysis of the individual phases of the new motion cycle has been performed by utilizing precise pneumatic actuation. Contactless magnetic actuation augments lobe motion into energy-efficient resonance-like oscillatory motion. The use of environmentally driven actuator materials has allowed autonomous motion generation via changes in environmental conditions. Measurement of the energy required for the differently actuated movements has shown that the Venus Flyflap is not only faster than the biological models in its closing movement, but also requires less energy in certain cases for the execution of this movement.

## 1. Introduction

Within the last decade, plant-inspired robotics has become established as a new emerging field of soft robotic science. One outstanding example is represented by the plant-inspired growing robots developed by the group at the Italian Institute of Technology; these robots have roots, tendrils, and leaves like their biological models and are able to grow, forage, and harvest energy from the environment [1,2,3,4,5]. A focus area within this field is the development of artificial systems with certain characteristics that are able (1) to exist autonomously, (2) to sense, adapt, and react to the environment, (3) to sustain their homeostasis by harvesting energy, (4) to sense damage, and (5) to possess self-repair functions. Such systems are currently still in their infancy but should revolutionize the technical world in the future.

The Venus flytrap (*Dionaea muscipula*) provides a suitable biological model for plant-inspired robotics. These plants perceive their environment and adapt and react to it, all of which are features required by robots. In addition, Venus flytraps carry out one of the most complex decentralized controlled process sequences in nature, namely a fast snap-buckling prey capture movement, which includes an upstream mechanical memory and several subsequent processes depending on successful prey capture [6]. The Venus flytrap can be considered as a bi-stable system, with the lobes being stable in a convex and concave configuration [6,7,8], changing through a snap-buckling motion similar to technical bistable systems [9,10,11,12]. The transference and combination of the basic principles of the Venus flytrap movement sequence has enabled the production of life-like artificial Venus flytraps (AVF) [13]. However, artificial Venus flytraps have so far been produced more as a by-product of the development of new actuation systems. AVFs have been used as suitable demonstrators of the capabilities of actuators in the development of novel pneumatic systems utilizing instabilities in elastic energy storage [14], magnetically driven bi-stable prepregs [15,16], ionic electroactive polymer metal composites (IPMCs) [17,18], light responsive liquid crystalline elastomers (LCEs) [19,20], and hygroscopic bistable systems (HBS) reacting to changes in humidity [21]. However, none of these AVF systems incorporates all the functions of the biological model *D. muscipula*. Most of them only exhibit a reasonably fast (sometimes without snapping) closure mechanism following an external stimulus, as Esser et al. [13] have shown in their review of the current state of AVFs. Nevertheless, a simple and low cost AVF with a novel motion pattern inspired by two carnivorous plants has been developed, inspired by those AVF systems: the Venus Flyflap (VFf). It uses various actuation modes and allows the investigation and characterization of the energy needed for actuation [22]. In addition, for the first time, this system combines motion features of two closely related snap-trapping carnivorous plants into one system, namely the snap-buckling of *D. muscipula* (Figure 1A,B) and the kinematic coupling and motion amplification of the Waterwheel plant (*Aldrovanda vesiculosa*) (Figure 1C,D) [6,7,23,24]. The trap closure movements based on these motion principles are among the fastest movements in the plant kingdom: *A. vesiculosa* needs 0.02 to 0.1 s [23] and *D. muscipula* 0.1 to 0.5 s [6] to close their trap lobes.

The compliant foil structure used in the VFf is based on a shape inspired by the biological models. A rectangle forms the base (analogous to the leaf midribs of *A. vesiculosa* and *D. muscipula*); attached to its long sides, two triangles represent the trap lobes, and two circles (‘ears’) at the short ends of the rectangle allow the actuation of the system through kinematic coupling (Figure 1E–G) [14]. A downwardly directed motion of the ‘ears’ results in motion amplification causing the fast closure of the system by a continuous motion, as found in *A. vesiculosa*. To be able to snap, the foil ‘midrib’ is reinforced with a plastic microscopic slide as a backbone. When force is applied centrally, the backbone changes its curvature, and the VFf opens its lobes in a fast snapping motion similar, but inverse in direction, to the snapping by curvature inversion in *D. muscipula*. Thus, a new motion sequence has been achieved by combining the motion principles of the two snap-trap plants into one artificial system. A fast (continuous) closing step, followed by a sudden release of the stored potential elastic energy in the system, causes a snap opening, as soon as the input energy exceeds a certain threshold.

The present study aims to answer the following questions. Is it possible to build a biomimetic system that combines and utilizes two movement principles in one cost-efficient structure, that mimics the biological models, and that can be triggered by environmental stimuli?

Direct actuation by means of pneumatic cushions enables the opening and closing movements of the VFf to be controlled in a targeted manner (Figure 1H). Thereby, specific aspects of the motion and system as a whole can be investigated separately. This should provide new insight into the motion behavior (closing and opening times and speeds), the kinematics, and the elastic energy storage of the system. Contactless actuation via a magnet varying the actuation speed is used to achieve and investigate a uniform energy-effective oscillation or resonance-like motion (Figure 1I). Concerning autonomous systems, the VFf can be also actuated by environmental stimuli such as heat and humidity or a combination of both, by using environmentally sensitive materials such as shape memory alloys and polymers in combination with hydrogels. Furthermore, a comparison of our system with the biological models and the state of the art in artificial Venus flytraps allows an evaluation of its biomimetic potential.

## 2. Materials and Methods

### 2.1. Setups for Movement Analysis of Dionaea Muscipula and the Artificial Venus Flyflap (VFf)

In order to enable a valid comparison of the motion characteristics of the *D. muscipula* plant with the pneumatically and magnetically actuated VFf, a kinematic analysis was performed using a video chamber fitted with two 1000 fps high-speed cameras (Baumer matrix monochrome camera VCXU 13 M/Imaging Solutions Motion traveller 1000) (Figure 2A). Videos were recorded at a constant frame-rate of 1000 fps with a resolution of 512 × 512 pix, and the recordings were synchronized using NorPix-StreamPix 8.0.0.0 (x64) Software. During actuation, the path of motion, movement speed, and kinematic parameters (speed and acceleration) were tracked and analyzed using the open-source software Kinovea (version 0.9.1). During the testing period of three weeks, 28 traps from four different *D. muscipula* plants were tested repeatedly. The tests were performed on three different dates with one week of “rest” in between to minimize the stress for the plants. Tracking markers were applied to the lobe tips of the pneumatic VFf and the magnetic VFf and to the magnets. The video and picture-based analysis of the thermally driven VFf actuated with shape memory alloy (SMA) springs (Figure 2B) and of the hydrogel actuated VFf (Figure 2C) required only a lower framerate because of the lower motion speed. Therefore, a digital camera (Panasonic Lumix DMC-FZ1000, Figure 2(C3)) with a recording framerate of 25 fps was used. In the case of the thermally driven VFf, the lobe tips and the spring length were tracked. Statistical data analysis was performed with the open-source software RStudio (version 1.2.5042).

The setup of the actuator systems is described in detail in Esser et al. [14] and shown in Figure 1. Further information is provided in the Appendix A section “Materials and Methods: 2.1 Standardized production of the compliant foil demonstrators”.

### 2.2. Energy Measurements

The energy consumption in each experiment was determined by measuring the amount of electricity that was required for actuation. The electricity consumption was determined with an energy-measuring device that was plugged into the socket before the actual electricity consumer, e.g., the pneumatic test bench, magnetic stirrer, temperature chamber, or steam source. Force-displacement measurements were performed for calculating the necessary work and kinetic energy to drive and trigger the closing and opening movements of the demonstrators (Figure 2F). A specific testing setup to apply force according to the actuation scenario was designed for each VFf system (Figure 2E). The pneumatic VFfs were fixed in an upside-down orientation in the testing machine, and the compression pistons applied force to the backbone through the casing openings for the pneumatic cushions (Figure 2(E3)). In the case of the magnetic VFf, a downward force was applied to the magnets until the lobes closed (Figure 2(E2)) Clamps for the tensile tests held the SMA spring of the thermal VFf inside the thermal chamber of the testing machine. The temperature was increased up to 65 °C, and the output force was measured. All measurements were performed with an Inspect universal testing machine (Hegewald and Peschke Meß-und Prüftechnik GmbH, Nossen, Germany). During all hysteresis measurements, the stroke was adjusted accordingly to achieve a full closing or opening movement. The movement of the machine caused further increasing forces after a complete opening, since the testing machine did not immediately change the direction of movement after opening. These measurement ranges were excluded from the kinetic energy calculations. Since the course of the force was not uniform, it had to be integrated over the entire path until the moment when both trap lobes opened (*s*_2_) (Figure 2F). The area under the force-displacement curve represents the work exerted on the moving object. By measuring the area, the work and thus the required kinetic energy can be calculated by:(1)$$W=\int_{s_{1}}^{s_{2}}\vec{F}\left(\vec{s}\right)*d\vec{s}.$$
with *W* = work/kinetic energy; *F* = force; *s* = stroke; *s*_1_ and *s*_2_ indicate the lower and upper end of the integral, respectively. The efficiency calculation and corresponding equations are shown in the Appendix A section “Materials and Methods: 9. Actuation system efficiency calculation”.

### 2.3. Motion Analysis of Discrete Repetitive Motion Generation by Pneumatic Actuation

Analysis of the complex motion pattern of the VFf was achieved by using directly driven pneumatic actuation of the opening and closing motions, which could be triggered separately. The foil-based VFf was attached to a 3D-printed (Rigur^TM^; Stratasys Ltd., Eden Prairie, MN, USA) ridged case that housed three pneumatic cushions (EcoFlex 0030; KauPo Plankenhorn e.K., Spaichingen, Germany) (Appendix A). Pressurization of the two smaller outside cushions pushed the rigid backbone of the VFf upwards, because its ears were attached to the casing (Figure 1H). This closed the lobes via kinematic amplification, as seen in *A. vesiculosa*. To trigger the opening movement, the central cushion inflated, while the outside cushions deflated. This caused the backbone to bend, leading to a fast “snapping” opening movement accompanied by an inversion of the spatial curvature of the lobes, similar to the snap-buckling principle of *D. muscipula*. Pressurized air was supplied by a pneumatic test bench [25] that enabled measurement and manipulation of the pressure and pressurization time for the closing and opening motions, respectively (see Appendix A for actuation pattern). As pressurization inflated the cushions and actuated the system, we further refer to the duration of pressurization as the actuation time (AT). The actuation times chosen for the characterization were 200 ms, 300 ms, and 400 ms, as the closing time of the biological model was below 500 ms. During actuation, the valves opened depending on the AT pressurizing the cushions. The actuation pressure was determined to be 0.7 bar in preliminary tests. This was the maximum pressure at which complete closure was guaranteed without the cushions bursting. The system pressure was adjusted accordingly to the AT in order to achieve a pressure of 0.7 bar within the actuators during each AT.

### 2.4. Motion Analysis of Contactless Actuation of the Demonstrator by a Rotating Magnetic Field

In the magnetic VFf, the foil demonstrator was directly attached to a 3D-printed casing without the rigid microscopy slide backbone. The casing in turn was attached to an aluminum profile in the video chamber (see Figure 1I and Appendix A). A set of two round flat magnets (Neodymium, 15 × 2 mm, 30 g each, 6 kg holding force) was attached to one of the ears. The VFf was placed 20 mm above a magnetic stirrer (IKA RCT B 5000) with adjustable rotations per minute (rpm) ranging from 100 to 1500 rotations per minute. The rotating magnet pulled and pushed the VFfs magnets, resulting in upwards and downwards movements of the VFf ear, which in turn set the lobes into a flapping motion. To investigate the possibility of reaching the natural frequency and gaining a resonance effect in the oscillating flapping motion of the system, six different actuation speeds were investigated with 400, 700, 800, 900, 1000, and 1300 rpm, respectively. In preliminary tests, the rpm were steadily increased from 0 to 1500 and the behavior of the VFf was observed. From 400 rpm onwards, the first clear movements occurred, at 800 and 900 rpm a uniform movement behavior was observed and for frequencies above 1300 rpm only irregular behavior. Because of this, the focus of the experiments was on the range of 800 and 900 rpm, and additional frequencies 100 rpm higher and lower were investigated. In addition, the extremes of 400 rpm (first clear movements) and 1300 rpm (only irregular erratic motions) were investigated.

### 2.5. Environmentally Triggerable Systems

#### 2.5.1. Motion Analysis of Thermally Actuated VFf by Using SMA Springs

To introduce autonomy into the system, the VFf was equipped with a temperature-sensitive SMA spring (Nitinol). The critical temperature for the SMA springs to induce contraction was determined by placing five springs in the temperature chamber of the universal testing machine at increasing temperatures, starting at 22 °C, and the length was measured each time that the temperature rose by 2 °C (see Appendix A and Appendix A). An SMA spring was attached to the foil demonstrator by using rivets on each lobe (see Figure 1J). The spring was stretched to a fixed length of 115 mm and then attached to the VFf. Three VFf were tested simultaneously in a climate chamber (Environmental test chamber CTC256, Memmert GmbH + Co. KG) that allowed programmed temperature settings (Appendix A). Three different temperatures were tested (55 °C, 60 °C, 65 °C) (Appendix A). The rising ambient temperature induced a phase transformation in the spring material reversing the deformation of the spring, which then shortened and, hence pulled the ears downwards closing the VFf.

#### 2.5.2. Motion Analysis in VFfs Actuated by Combination of Two Stimuli: Humidity and Temperature

By using a hydrogel-coated 3D-printed shape memory polymer (SMP), an actuator system was created that responded to a combination of two environmental stimuli. The hydrophilic terpolymer (hydrogel) at a concentration of 100 mg/mL was applied manually to the SMP midrib backbones, which had a low T_G_ of 50 °C (Rigur^TM^; Stratasys Ltd., Eden Prairie, MN, USA) (Figure 1K). After a drying step at 65 °C, the terpolymer was simultaneously cross-linked and surface-attached by UV irradiation at 365 nm. During drying, the surface-attached hydrogel shrank leading to a bent backbone (see Appendix A). This process thus formed a water-sensitive hydrogel network that actuated the demonstrator when exposed to higher (or different) levels of humidity. The bending angle was measured via ImageJ (version 1.53a, [26] (see Appendix A). After being crosslinked, the bent backbone was attached to a snapped-open foil VFf. The VFf full assembly (Appendix A) is described in more detail in the Appendix A section “8. Environmentally triggerable systems: Stimulus combination humidity and temperature”.

In the fully assembled VFf testing scenario, the change in environmental conditions was achieved through a source of hot water vapor (REER FD 1540) that generated a constant flow of hot steam. The VFfs were positioned in the steam at a fixed distance of 5.7 cm to the outlet for 300 s. This caused hydration of the dried hydrogel and led to swelling, which reduced the bend in the now flexible backbone (heated above the T_g_ by the hot steam). This unlocked the snapped-open VFf, the stored elastic energy was released, and the lobes moved upwards to the resting position. Thereafter, the VFf was ready for manual actuation. Nineteen backbones were tested repeatedly for five dehydration and rehydration cycles. For the identification of the threshold humidity at which the hydrogel-coated backbones straightened, ten were tested in a small climate chamber in which the humidity was raised from 20% to 80% at a constant temperature of 65 °C.

### 2.6. Statistics

The open-source software RStudio (v. 1.2.5042, R Core Team 2017) and statistic programming language R were used for statistical analysis and calculations. For the pneumatic demonstrator, we used a two-way ANOVA on ranked transformed data, having checked for normal distribution (Shapiro–Wilk test) and homoscedasticity (Bartlett test). The significance levels and correlation between variable actuation times and movement speeds were determined. Post-hoc tests were performed via multiple comparisons by using Tukey’s test. Data from the kinematic analysis of the magnetic demonstrator were checked for normal distribution (Shapiro–Wilk test) and homoscedasticity (Bartlett test). Significant differences between movement speed and actuation rpm were determined using the Wilcoxon signed-rank test. For the hydrogel- and SMA-based demonstrator, we used a paired Wilcoxon signed-rank test, having checked for normal distribution (Shapiro–Wilk test) and variance homogeneity (Kruskal–Wallis test).

## 3. Results

### 3.1. Movement Characteristics of the Biological Model D. muscipula

The movement of the trap leaves of the biological model *D. muscipula* was observed from two perspectives (Figure 3A), and the leaf rim was tracked in order to calculate movement speed (Figure 3B). The closing time of the five tested plants ranged from 148 ms to 1821 ms with a mean value of 599.1 ms (SD = 487.7 ms). During the three-week testing period, mean closing time increased each week of testing from 549 ms in the first week to 683.9 ms in the third week (Figure 3C). Of the closing events, 56.1% took place in less than 500 ms (Figure 3D). Closing times above 1000 ms were the result of a delay between the initiations of the closing motions of the two lobes (Figure 3). Speed values of the closing motion ranged from 0.016 m/s to 0.245 m/s with an average movement speed of 0.088 m/s. The corresponding raw data is shown in Appendix A in the Appendix A section “Materials and Methods: 3. Kinematic analysis of the biological role model”. 

### 3.2. Motion Analysis of Generation of Discrete Repetitive Motion by Pneumatic Actuation

The kinematic analysis of three pneumatically actuated VFfs showed that a movement cycle consisting of one closing and opening movement could be separated into five separate phases (Figure 4A), with a curvature inversion and unrolling movement of the lobes during the snap opening of the VFfs (Figure 4B). During actuation, the highest velocities were observed during the opening movement (Figure 4C). The closing movement lasted between 132 ms and 311 ms (Figure 4D). The opening time ranged from 19 ms to 56 ms, depending on the time delay between the opening events of the two lobes. The fastest opening times were observed when both lobes moved simultaneously in parallel. Opening time is defined as the time needed from the first sign of the opening movement to the passing of the horizontal line of the backbone. After opening, with a still-inflated central cushion, the demonstrator remained locked in the open position. When all pressure was released from the system, the VFf returned to its initial position. The closing velocity showed a dependence on actuation time (AT) (a longer AT of 200, 300, and 400 ms gave slower closing) (Figure 4E). In contrast, the AT had almost no significant effect on the opening velocity of each lobe (Figure 4F). The measured speeds of the lobe tips during opening ranged from 3.05 m/s to 4.91 m/s (Figure 4F). Demonstrator three showed faster opening times because of the simultaneous opening of the two lobes (Figure 4F). The applied pressure ranged between 0.75 and 0.79 bar for the outside cushions and 0.66 and 0.69 bar for the single central pneumatic cushion.

### 3.3. Motion Analysis of Contactless Demonstrator Actuation by a Rotating Magnetic Field

During actuation at low speed (rounds per minute) of the magnetic stirrer, the VFf lobes showed shivering movements, while the permanent magnet moved up and down depending on the speed of the magnetic stirrer (Figure 5A). The motion accelerated in association with an increase in actuation velocity, transforming into a fast, nearly regular, oscillating flapping motion around 800–900 rpm with a stable motion frequency of 15.6 Hz at 900 rpm (Appendix A and Appendix A). At 1000 rpm, the bent lobes tended to buckle and quickly returned to their original shape thereafter (Figure 5C); the highest observed velocity at this time was 3.56 m/s (Figure 5B). The average speed of lobe tips increased with rising actuation velocity ranging from 0.33 m/s to 2.37 m/s. In correspondence the motion frequency also increased from 3.67 Hz to 44 Hz, with an observed decrease as the motion became more erratic at higher rpm (Appendix A and Appendix A). The movement of the lobes was delayed compared with the perpendicular motion of the magnet (Figure 5C). 

### 3.4. Environmentally Triggerable Systems

#### 3.4.1. Thermally Driven VFf

When the surrounding air is heated above the critical temperature of the SMA spring, temperature-induced phase transformation takes place, and the spring contracts and generates enough force to pull the ears downwards thereby closing the lobes. To ascertain the critical temperature of the SMA springs, they were stretched to 115 mm and placed in an oven. Between 60 °C and 65 °C, four of the five tested springs reduced their length by at least 50%. The spring that did not contract sufficiently was not tested further (see Appendix A). In some cases, these differences may originate from the production process, as these were commercially available SMA spring. Three springs with comparable contraction behavior were selected for further experiments with the VFf. These were attached to the VFf ears, and the closing behavior was investigated at three different temperatures (55 °C, 60 °C, and 65 °C) via video analysis and motion tracking (Figure 6A,D). No movement was detected at 55 °C, only one demonstrator closed its lobes at 60 °C, whereas all demonstrators performed closing movements at 65 °C (Figure 6B). The closing movement was triggered by a reduction in spring length, which was here measured at the beginning and the end of a heating cycle at the target temperature (Figure 6D lower orange lines). Spring contractions ranged from 0 cm to 5.44 cm (Figure 6B). A minimal spring length reduction of 0.56 cm was needed fully to close the lobes of the demonstrator (Figure 6B red line). The overall time needed from the first sign of movement to the complete closure of the VFf was defined as the closing time. The closing times of the three VFf ranged from 36 to 429 s with an average of 234.7 s (Figure 6C), the fastest closing times being observed at 65 °C. The average closing speed ranged from 0.19 × 10^−3^ to 1.1 × 10^−3^ m/s (Figure 6E). The force generated by the spring contraction was measured at 65 °C. Spring two and three showed no significant difference with an average force between 4.17 N and 6.83 N (Figure 6F). The corresponding raw data for spring length change is shown in Appendix A and for demonstrator closing times in Appendix A in the Appendix A section “Materials and Methods: 7. Environmentally triggerable systems: Thermally driven AVF using shape memory ally (SMA) springs”.

#### 3.4.2. Motion Analysis in VFfs Actuated by a Combination of Two Stimuli: Humidity and Temperature

The change in curvature of the hydrogel-coated backbones under changing environmental conditions was analyzed in a small climatic chamber that allowed controlled changes to be made in air humidity and temperature. Seven hydrogel-coated backbones were tested at a constant temperature of 65 °C, and the bending angle was observed at various humidity levels (Figure 7A). All seven backbones straightened completely at 60% relative humidity. The process started slowly, and minor changes became visible at 25% to 35% humidity. Between 35% and 60%, the decrease in the opening angle accelerated, levelling out at 65% to 80% humidity (Figure 7B).

After attachment of the hydrogel coated backbone to the foil VFf, a higher environmental humidity of about 70% was needed to reduce the curvature of the backbone and thus to unlock the demonstrator. At higher humidity, the hydrogel swelled to a greater extent, as was necessary to overcome the resistance of the foil material. After the VFf had been placed in a stream of hot steam (Figure 7C), it took an average of 64.5 s to unlock and reach the initial resting position for manual actuation (Figure 7D). Out of 95 tests, 62 demonstrators unlocked correctly during the 300 s rehydration time in hot steam. The average time required from the first contact of the steam to the unlocking of the VFf varied from 33.2 s to 126 s during the five cycles, whereas the actual unlocking movement (*n* = 19) required approximately 0.404 s (sd = 81.4 ms).

### 3.5. Energy and Work/Kinetic Energy Requirements and Stored Energy for and during Lobe Movement

The energy consumption of each experiment was determined by measuring the amount of electricity that was required for actuation. The electricity consumption was determined with an energy-measuring device that was plugged into the socket before the actual electricity consumer. Between 0.24 and 0.23 L of compressed air was needed for each actuation cycle of the pneumatic demonstrator, and the pneumatic test bench consumed 1 J per cycle (closing and opening). The magnetic stirrer consumed between 27 J and 39 J for the duration of actuation depending on rotation speed. Unlocking of the heat- and moisture-driven hydrogel actuator through a hot water steam source consumed 129,712 J, whereas the SMA-based actuation through contactless heating in a temperature chamber consumed 5,464,800 J of electric energy (Table 1). The efficiency of the systems is for pneumatic actuation around 3.7%, for magnetic actuation 5.6%, whereas for the SMA spring it is far lower as the temperature chamber used has a very high power consumption (Table 1).

Force-displacement measurements were performed to determine the amount of kinetic energy (work) required either to open or to close the VFfs (Figure 8). The closing movement of the pneumatic VFf needed an average amount of kinetic energy between 19.04 mJ and 30.81 mJ. Most of the time (97.3%), the opening motion of the two lobes was slightly time-shifted, as one lobe opened before the other (Figure 8C,D). For the opening movement of the VFf, an average kinetic energy requirement of 24.42 mJ to 34.65 mJ was needed to trigger the first opening event (Figure 8D,E). Between 38.49 mJ and 79.54 mJ were necessary for the second opening event (Figure 8D,E). A parallel or simultaneous lobe opening demanded less kinetic energy (around 37 mJ), although the force needed was higher (10 N compared with 9 N for the second lobe opening). The overall stroke necessary to achieve the opening movement was less than that for the non-parallel opening (Figure 8F). During opening, this difference was also reflected in the observed force drop, which was lower for non-parallel opening than for parallel simultaneous opening (Figure 8E). The release of the stored energy was reflected in the drop in force after the opening of a lobe. The released stored energy was calculated with Equation (1) to be approximately 0.6 mJ for the opening of one lobe and 2.4 mJ for the opening of both lobes in parallel.

Similar experiments were performed on the magnetic VFfs by applying a downward force on the magnets (Figure 8G,H). The mean kinetic energy requirements ranged between 11.63 mJ and 17.31 mJ. The measured curves for the closing movement of the pneumatically and magnetically driven VFf differed from each other. Both showed a linear increase in force at first, but for the pneumatic demonstrator, the curve became steeper at the end of stroke (B), whereas for the magnetic demonstrator, it flattened, and even decreased (H). The average force generated by the SMA springs at 65 °C ranged from 4.17 N to 6.83 N (Figure 6F) and the change in spring length (Figure 6B) from 5.6 mm to 54.4 mm. Based on these values, the average kinetic energy requirements computed to 48.1 mJ to 315.9 mJ. The energy values of all tested VFf systems in comparison with the biological model are shown in Table 1.

## 4. Discussion

We have created an environmentally triggerable biomimetic system combining two movement principles in one cost-efficient structure mimicking biological models. The movement characterization with direct and precise pneumatic actuation has revealed that one motion cycle of our VFf system consists of five phases, beginning with a fast closure within 311 ms, followed by an even faster snap opening within 73 ms based on the release of stored elastic energy during opening. Through contactless, magnetically driven actuation, the VFf achieves an oscillating motion of the lobes near to the estimated resonance frequency of the system. The SMA spring and hydrogel-coated SMP backbone actuated VFfs highlight the possibility of an actuation of the system induced by environmental stimuli. Overall, the first combination of two snap-trap motion principles within one compliant system provides the baseline not only for novel artificial Venus flytraps, which are as fast as the biological model (pneumatic VFfs) and which act just as autonomously (3D-printed hydrogel-coated VFfs), but also for application in the fields of bioinspired architecture, autonomous systems, and soft robotic actuation.

High-speed recordings have allowed us to study the complete movement of the capture lobe of *D. muscipula*, an event that lasts on average less than 500 ms, matching well with the literature values (100–500 ms [6,27,29]). We have observed that, during a three-week testing period, the plants tend to take longer and longer to close after an unsuccessful attempt at catching prey, a rest period, and a full reopening. We have been able to record and track the time and speed of the fast snap buckling motions in the biological model and to compare them with measurements of the motions of our VFf systems. With the presented characterization process and setup, it is possible to analyze fast 3D movements with high-resolution videos at 1000 fps frame rate.

### 4.1. Comparison with the Biological Model

Our VFf systems take less time for closure and generate higher speeds than the biological models (Table 1). Table 1 represents an updated extension of the table published by Esser et al. [13] and compares the individual VFf systems of this study with the biological model. To determine the energy required for closure, the energy consumption of each experiment was determined by measuring the amount of electricity that was required for actuation via an energy-measuring device. Force-displacement measurements were performed to determine the kinetic energy required for closure and the potential energy stored in the system during closure. The VFf systems required less energy (1 J in the case of pneumatic actuation derived from electrical energy consumption) for lobe closure than *D. muscipula*, which needed approximately 300 µmol ATP (ca. 10 J under standard conditions [27,28]). A comparison with other AVF literature values is difficult, since no energy measurements or calculations have been performed for most systems. The SMA spring-driven system of Kim et al. [30] needs 12.4 J for closing and 48 J for reopening, which are lower values than for our thermally driven SMA system and higher for the biological model. Their AVF system is faster in closing than our SMA-driven system. As we use a thermal chamber for heating instead of direct heating via electrical current (joule heating), our system heats up far more slowly than that of Kim et al. [30]. However, the former more clearly reflects heating attributable to a change in ambient temperature and validates the feasibility of our system for its intended use. In contrast to the system of Kim et al., the highest movement speeds in our study were achieved with the pneumatically and magnetically driven VFf systems.

### 4.2. Combination of Two Snap-Trap Principles Gives a Novel Pneumatically Driven Motion Sequence

The actuation time (AT) significantly influenced the closing time of the pneumatic VFf, with a shorter AT resulting in a shorter closure time. As the necessary actuation pressure inside the cushions was 0.7 bar, the overall system pressure had to be higher to achieve this value during shorter AT. The closed state of the pneumatic VFf represents a high energetic state, where energy is stored by elastic deformation within the backbone and by distortion of the lobe geometry. This energy can only be released suddenly by passing a threshold in pressure and increasing the curvature of the backbone, which thereby transforms the stored elastic energy into kinetic energy causing the observed rapid movement of the lobes. Thus, the pneumatic cushions are not involved in this opening process; as such, their AT has only marginal to no influence on the opening time. The small differences observed are attributable to variations in the acceleration of the lobe. Significant differences above 40 ms in the opening times are attributable to a delayed and non-parallel opening movement of the two lobes. The observed large differences are probably based on inaccuracies during the manufacturing process, as the demonstrators are produced by hand, which can lead to (small) geometric deviations between the individual demonstrators.

A comparable pneumatic AVF system from Pal et al. [14] consists entirely of silicone and is able to close its trap lobes within 50 ms when using 0.35 to 0.7 bar for actuation, comparable with the pressure used in our pneumatic VFfs. In contrast to Pal’s system, our system stores energy during the closure movement within its lobe curvature similar to *D. muscipula* and does not rely on a pre-stretched layer. Therefore, the opening of our system shows actual snap buckling similar to that achieved in the biological model by releasing the stored energy in the lobes and thereby producing a rapid movement.

### 4.3. Resonance-like Movement and Generation of Contactless Fast Flapping Motion

Driven via a rotating magnetic field, the VFf is able to achieve a uniform oscillation or resonance-like motion depending on the rotational speed. Since the lobe motion is kinematically coupled to the ear motion, a significant influence of the different actuation speeds on the lobe motion in terms of velocity and overall motion has been observed. At 400 and 700 rpm, only minor motion of the VFfs is observed, but as the system approaches its presumed natural frequency at about 800–900 rpm based on acoustic and visual regularity, motion becomes more regular, and high flapping velocities and large amplitudes of motion are achieved. At more than 900 rpm, the movement becomes more irregular, and the lobes undergo kinking. Their movement could no longer follow the movement of the ear, which moves up and down faster than the lobes.

A comparison with the magnetic AVF from Zhang et al. [15,16] is difficult as their system uses an electromagnet to repulse a magnet attached to the AVF lobe, closing the system in 100 ms. Our system does not fully close its lobes and performs a flapping motion instead of closure.

### 4.4. Environmentally Triggered Motion

The closing of the SMA spring was significantly slower compared with that of the pneumatic VFf presented. Once the transition temperature was reached, the movement was continuous but slow. Notably, the used climate chamber showed effects of inevitable thermal stratification because of the low fan speed used during video recording (Appendix A). This led to the VFf nearest to the ceiling of the chamber reaching the critical transition temperature first. The systems showed no significant difference in closing speed or time (necessary for full closure) once the closing motion was initiated at transition temperature.

A direct comparison with the SMA-powered AVFs by Kim et al. (2014) is impossible, as the latter is based on bi-stable prepreg surfaces, in which the SMA springs are used to overcome the tipping point in the bi-stable system resulting in a fast-snapping motion, which takes from 0.4 s to 1.2 s [30].

Autonomy and environmental triggers are introduced into the system with the incorporation of shape memory materials. The VFf closure movement can be triggered by a change in temperature. This allows the system to be locked into a snapped open state until a specific stimulus combination unlocks the system and triggers the release of stored elastic energy. The blocking and locking of the movement and the specific release mechanism are achieved by the material properties of the components and the combination of shape memory polymers with hydrogels. We suspect that the observed differences between the individual test runs are attributable to the reduction and build-up of internal stresses in the system. The test conditions were not changed between the test runs, and the demonstrators were all dried in the same way. This creates an unwanted variance within the system, an issue that will be investigated in further studies. A comparable hydrogel-based system is the hydroscopic bi-stable sheet AVF of Lunni et al. [21], which responds to a 30% increase in relative humidity. Our hydrogel requires an increase of 35% in combination with the SMP backbone to create movement.

### 4.5. Overall System Discussion and Outlook

The novel motion sequence achieved here by combining the movement principles of two snap-trapping plants uses motion amplification for closure and snap buckling for opening. The closure movement temporary stores energy via the deformation of the foil and backbone: through the bending of the attached backbone, the foil crease buckles, and the systems snaps open. This novel motion pattern can be used for various applications in energy harvesting, safety switches, escape movements of UAVs and bioinspired architecture. Triggering and actuating the motion can be achieved through various actuation principles.

Compared with the biological model and other AVF systems, only the pneumatically actuated VFf requires less energy for actuation and shows a similar kinematic behavior to the biological model with curvature reversal of the lobes and a fast snapping motion. Moreover, the system is easy and inexpensive to produce, by using office supplies (overhead foil, magnets, double-sided adhesive tape) and 3D-printed parts with a low heat deflection temperature in the range of 50 °C. Our systems provide an alternative to the current state-of-the-art AVF in terms of actuation and environmental sensitivity and allow various types of actuations to be tested in the same base module.

For example, the pneumatic actuation system represents a suitable actuator to achieve precise and controllable motions. In combination with the specific structure of the VFf, the novel movement and snapping mechanism builds a mechanical energy buffer. During the closure movement, potential energy is mechanically stored in the system and is explosively released during parallel opening. The attachment of triboelectric harvesters to the lobes might allow the partial recuperation of the kinetic energy of the snapping motion (release of potential energy during opening 2.4 mJ), similar to the leaf harvesters of Meder et al. [1,31].

By using different materials and material combinations, the principle shown can also be transferred to other application areas. The autonomous actuation depends on the properties of the materials employed and can be achieved by other environmentally triggered shape memory materials, e.g., liquid crystalline elastomers [20,32,33,34]. Important here is the geometry and the combination of flexible and rigid material, equivalent to those of the foil and backbone, which enable a snapping motion in the first place. The system can also be implemented using a multilayer film system, whose individual films have different coefficients of thermal expansion, strengths, and stiffness.

The combination of the present actuation strategies into one system is yet to be achieved. Two combination strategies are possible. First, the combination of pneumatic actuation and the hydrogel/SMP-based system initialization mechanism should enable quick opening and closing via pneumatics and the locking of the system in place by hydrogel drying without having to expend energy. The second possibility arises from combination of the thermally driven SMA spring with a hydrogel latch creating a latch-mediated spring actuation (LaMSA) system [11,35]. The latch holds the spring in place, until an increase occurs in humidity. To function, the hydrogel-coated backbone must be stiff enough to withstand the contraction force generated by the SMA spring. This only applies if the system is heated above 65 °C, e.g., by direct sunlight, which causes the spring to contract and the SMP backbone to heat up beyond its T_G_. In future studies, flexible sensors will be integrated into the system, representing a first step in the development of a fully autonomous AVF system.

## 5. Conclusions

In this study, we present a novel motion sequence for plant-inspired robotics that combines, for the first time, the motion principles of two snap-trap mechanisms, as observed in plants, into an artificial Venus flytrap, namely the artificial Venus Flyflap (VFf) system. This system is based on a compliant foil with a rigid backbone resembling the snap-traps of plants. A closing step via kinematic coupling and motion amplification of the lobes and a fast opening through snap buckling of the VFfs backbone characterize the motion. A uniform resonance-like motion is achieved via actuation through rotating magnetic fields. Two autonomous environmentally triggerable versions are presented based on shape memory materials in combination with a hydrogel coating that reacts to changes in temperature and humidity. In addition, a novel selective locking mechanism is introduced that only unlocks if two environmental stimuli (in this case, changes in humidity and temperature) are present at the same time. The VFf system is characterized in terms of motion kinematics, speed, times, and energy requirements and is compared with the biological model and state-of-the-art AVF systems, all of which highlight the biomimetic potential and low energy demand of our system.

In future studies, the presented environmentally triggered actuators will be merged into one system with self-healing foil materials systems. In addition, the integration of soft flexible sensors and flexible solar cells as energy harvesters will lead to an autonomous system with embodied intelligence and energy. Hereby, we should achieve an autonomous, adaptive, and self-healing artificial Venus flytrap system advancing soft plant-inspired robotics and bringing technology one step closer to the ingenuity found in naturally occurring systems.

## Figures and Tables

**Figure 1 biomimetics-07-00099-f001:**
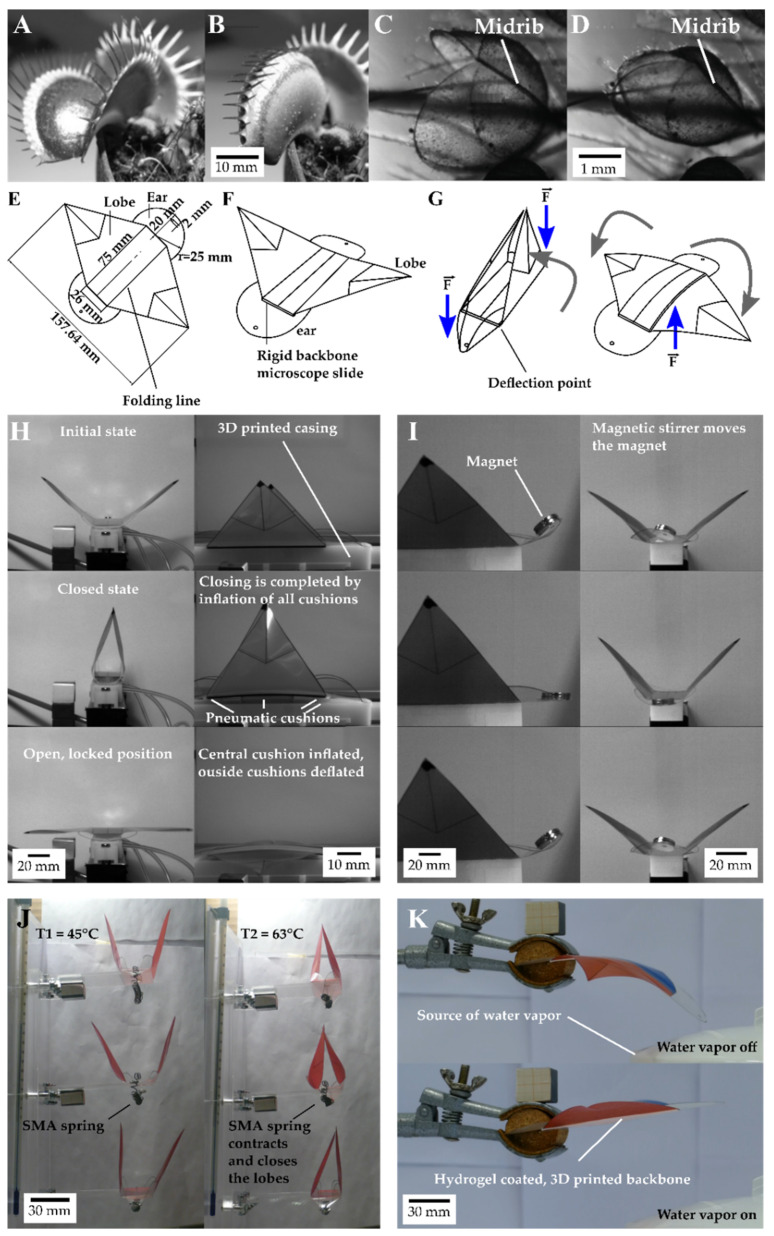
After being triggered, the Venus flytrap (*Dionaea muscipula*) closes its lobes by inverting the curvature of the lobes from, as viewed from the outside, concave (**A**) to convex (**B**) through the release of stored elastic energy. In the waterwheel plant (*Aldrovanda vesiculosa*) (**C**), water displacement in turgescent cells lying along the midrib combined with the release of stored elastic energy results in bending of the midrib, which is kinematical amplified and causes trap closure (**D**). The compliant foil demonstrator, the Venus Flyflap (VFf) (**E**), combines both mechanisms to close and open its artificial lobes. When force is applied to the ears (**F**,**G**), the lobes close via a kinematic coupling mechanism. Force applied to the backbone triggers a fast-snapping opening movement similar to the closing movement of the Venus flytrap. These mechanisms have been incorporated into various actuation scenarios. Pneumatic cushions drive the motion and enable the characterization of each phase of the motion sequence (**H**). A resonance-like, rapidly oscillating, flapping motion is achieved by a magnet that is attached to the ears and that responds to a rotating magnetic field (**I**). The VFf can also be actuated by environmental stimuli, when fitted with shape memory alloy (SMA) springs (**J**) or a combination of environmentally sensitive materials such as SMA and polymers in combination with hydrogels reacting only to a change in humidity and temperature (**K**).

**Figure 2 biomimetics-07-00099-f002:**
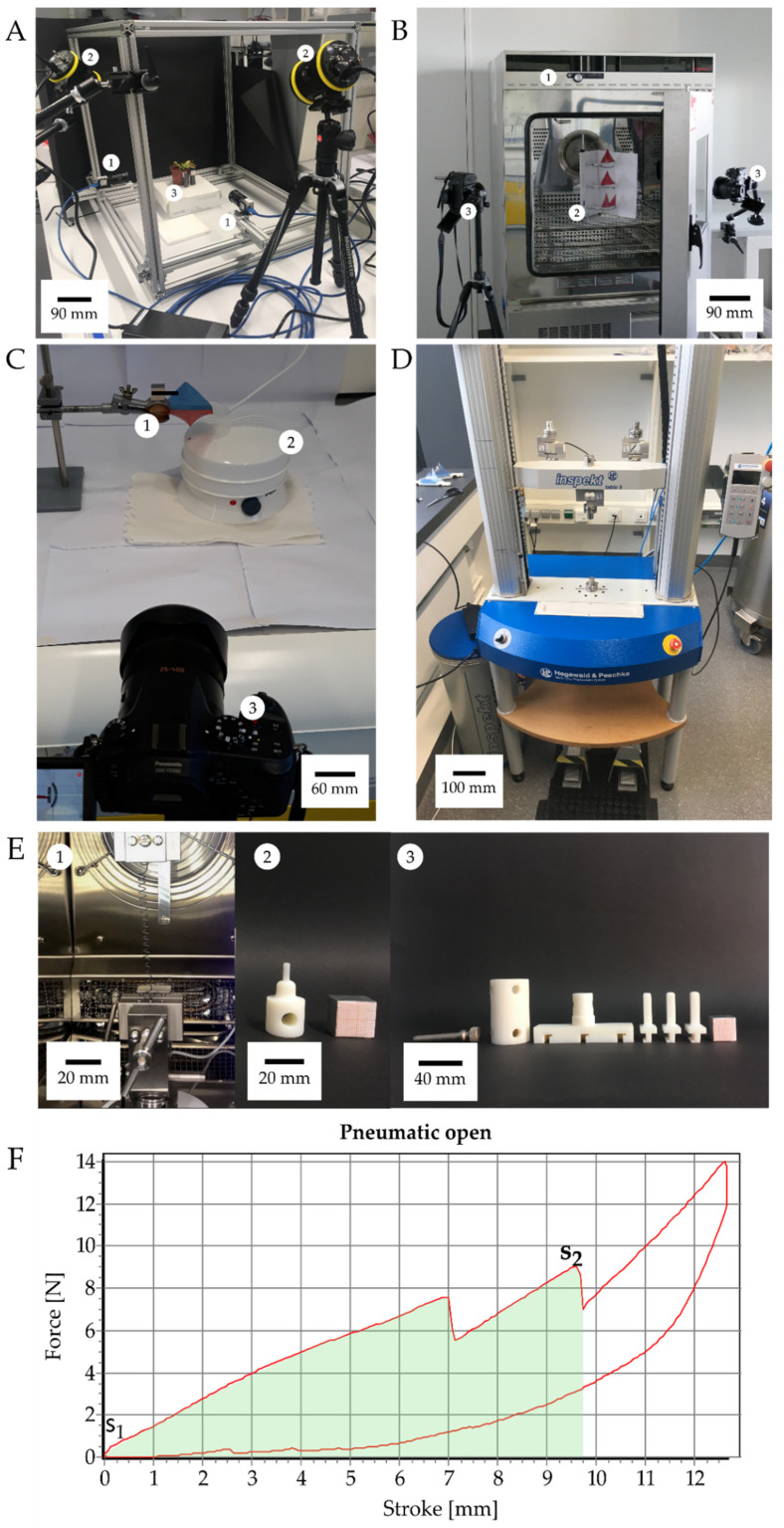
Characterization setups. High-speed videos were recorded in a specifically build recording chamber (**A**) with two highspeed cameras (**A1**) positioned at a 90° angle to each other. Light is provided by two high-power flicker-free LED light sources (**A2**) that point at the object of interest (**A3**). The Venus flytrap and the pneumatic and the magnetic demonstrators were recorded using this setup.The SMA actuated demonstrators were tested by placing them in a temperature chamber (**B1**). Three demonstrators were tested simultaneously (**B2**). The movement of the springs and the lobes of the demonstrator were recorded by two cameras (**B3**) for later analysis. For actuation of the hydrogel-based demonstrator (**C1**), a hot water vapour source (**C2**) provided enough humidity to unlock the demonstrator, which was filmed by a camera for kinematic analysis (**C3**). Force-displacement measurements were performed with specific test mounts by the Hegewald and Peschke Inspekt Table 5 (**D**). The output force for each SMA spring was measured (**E1**) at 63 °C. To mimic the actuation procedure, specific mounts were designed to apply pressure in the direction of actuation in the demonstrator (magnetic (**E2**) and pneumatic (**E3**)). An exemplary force displacement curve of an opening event of the pneumatic system is shown in (**F**). S_1_ and S_2_ indicate the lower and upper end, respectively, of the integral (green area) used for calculating the kinetic energy necessary to open the the two lobes (lobe openig indicated in the curve by the two sudden force reductions).

**Figure 3 biomimetics-07-00099-f003:**
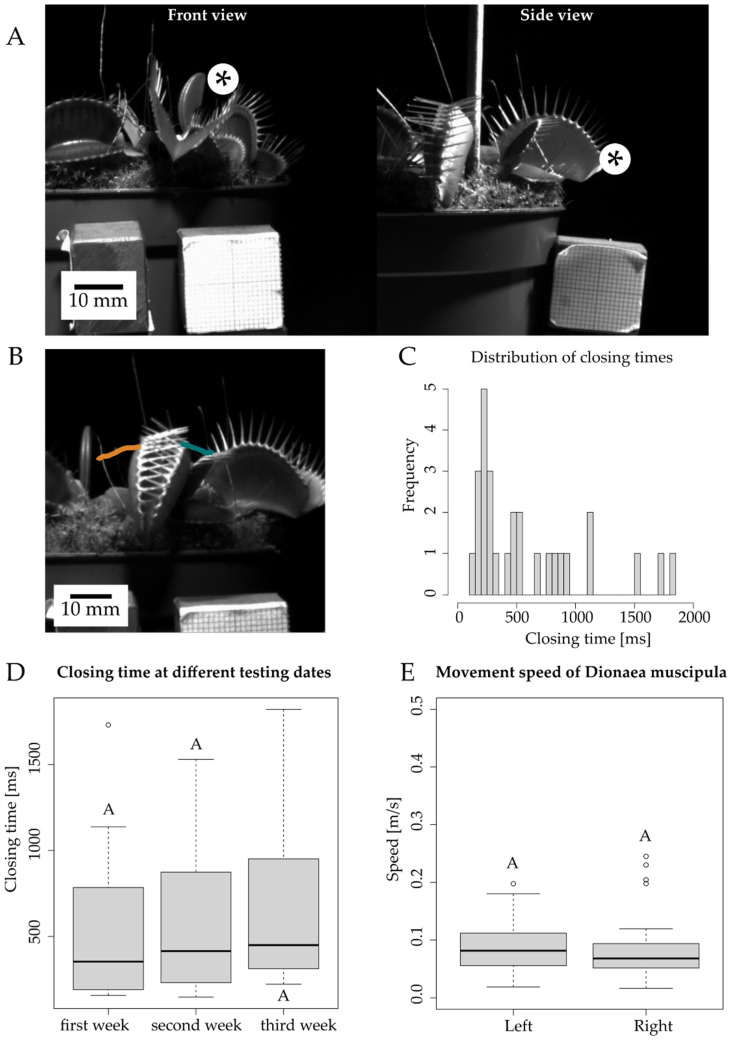
High-speed recording of the Venus flytrap from two angles gave a good impression of the overall movement (**A**). The path of motion was tracked at the base of the marginal teeth (orange and green lines) (**B**), and the movement speed was measured for these particular trajectories. Average closing time ranged from 550 to 680 ms for the three weeks of testing (**C**), whereas over half of the closing events needed less than 500 ms. The box plots represent 12, 8, and 8 measurements for week one, two, and three, respectively (**D**). Lobe tips reached velocities of between 0.016 and 0.24 m/s during closing. Each plot includes 28 measurements (**E**).

**Figure 4 biomimetics-07-00099-f004:**
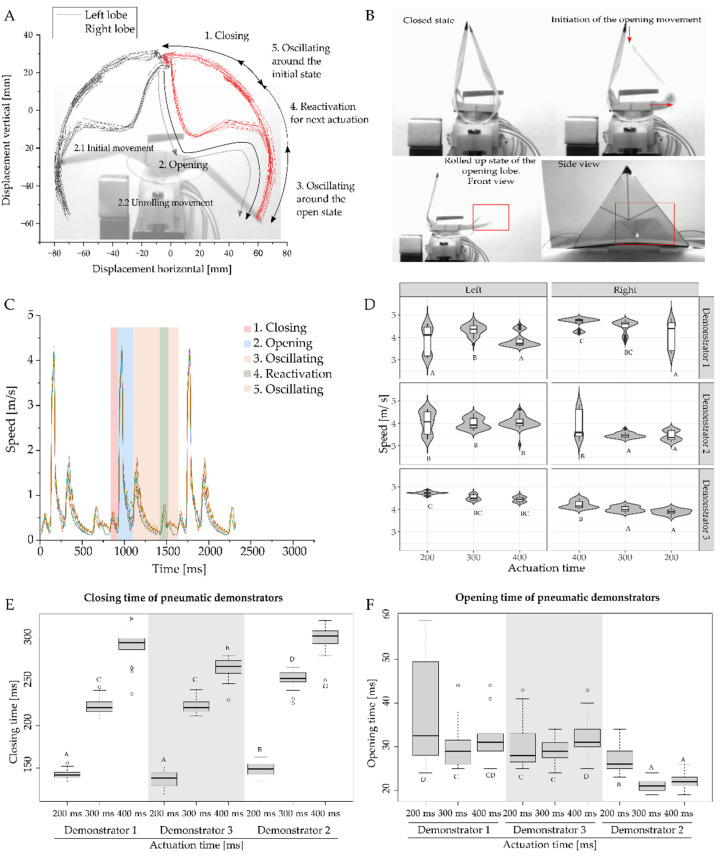
Kinematic analysis of the five phasic motions of the pneumatic VFf. After activation of the outside cushions, the VFf closes, and the opening movement is triggered when the central cushion is inflated. Examination of the lobe movement reveals five phases (**A**) starting with the closing movement (**A1**). The second phase is the opening movement (**A2**) that begins with a slight outward movement of the base of the lobe leading to a rolled-up intermediate state that is followed by a fast-unrolling movement (**B**). After opening, the lobes oscillate in the locked open position (**A3**). Following air release, the VFf snaps back into its original position and is ready for the next actuation (**A4**) with the lobes oscillating around the initial position (**A5**). In (**B**), a time-delayed opening of both lobes shows the unrolling movement of one of the lobes. The five phases are characterized by their different movement speeds (**C**), an example of which is shown here for a motion cycle at 400 ms AT. During actuation, the highest velocities (up to 4.9 m/s) are observed during the opening movement (**D**). Closing times depend on the overall actuation time (**E**). Significant differences above 40 ms for opening times are attributable to a delayed and non-parallel opening movement of the two lobes. Faster opening times represent simultaneous opening (**F**). Each boxplot corresponds to five measurements during which the three demonstrators have each completed eight movement cycles. Significance levels are indicated by the capital letters next to each boxplot. Different letters represent significant differences from other results with *p*-values below 0.05.

**Figure 5 biomimetics-07-00099-f005:**
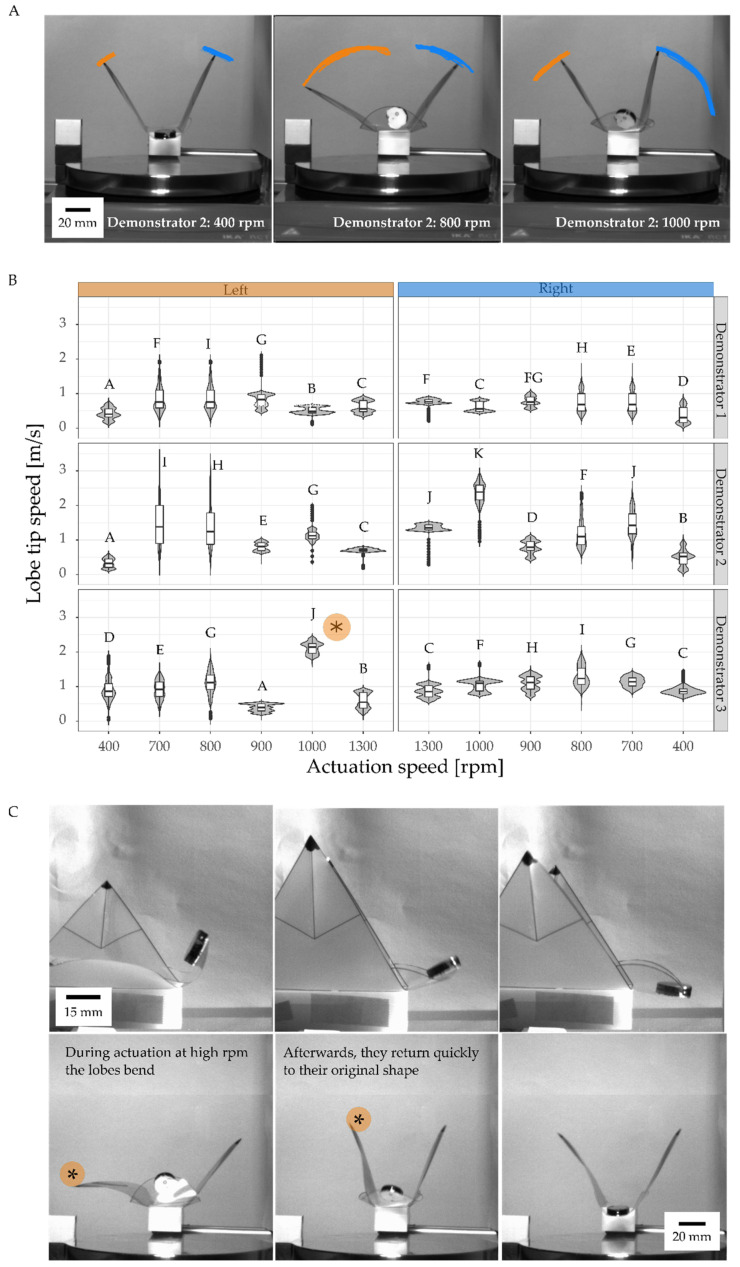
Motion analysis of magnetically driven oscillating VFfs with velocities approximating to natural frequency. With higher actuation velocity (higher rpm), the movement speed and amplitude increase (motion tracks indicated by orange and blue lines) (**A**). Three VFfs were tested at six different actuation speeds (**B**). Significance levels are indicated by the capital letters next to each boxplot. Different letters represent significant differences from other results with *p*-values below 0.05. Recording frame-rate of 1000 fps for a video duration of 3 s resulted in 3000 values for each speed (*n* = 3000) of the movement of the three tested VFfs. Highest velocities were observed when a lobe bent during the outward and downward movement and then snapped back into its original curvature (**C**). The images illustrate the difference in speed of the left and the right lobe of demonstrator 3 at 1000 rpm. * indicates the fastest snapping lobe in (**B**,**C**).

**Figure 6 biomimetics-07-00099-f006:**
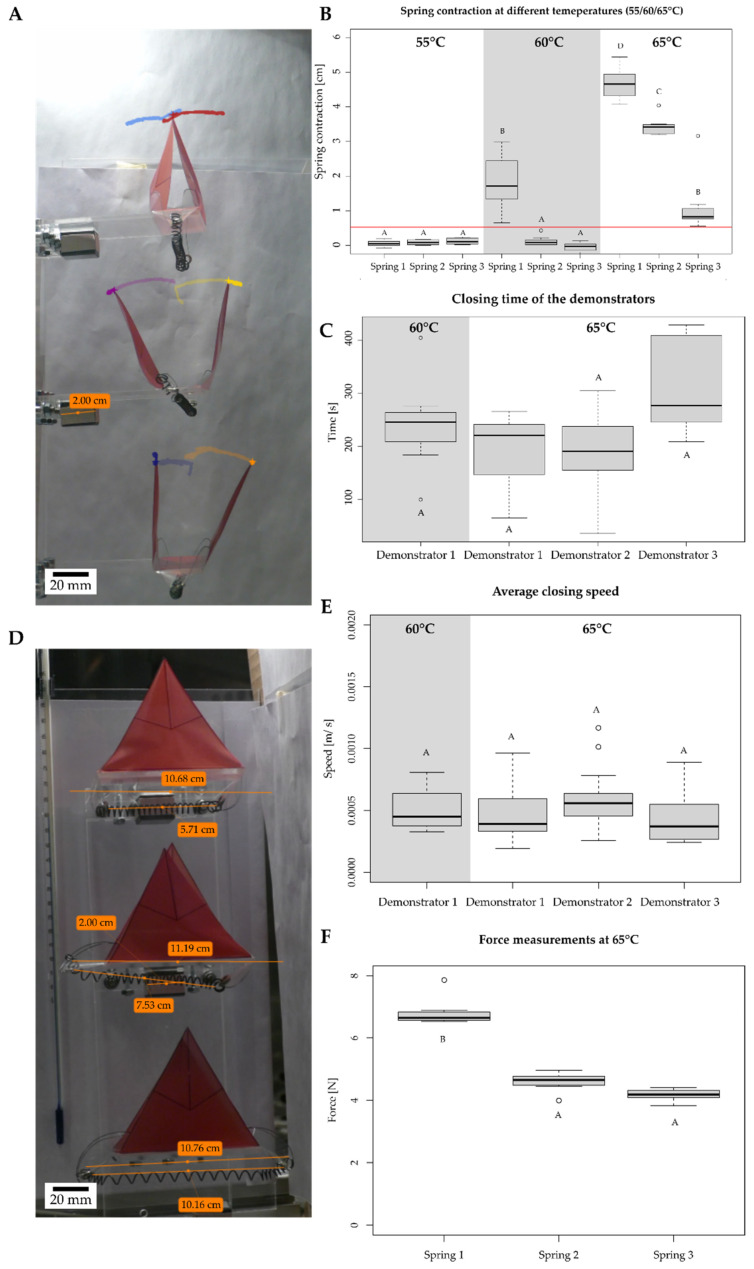
Kinematic motion analysis of the thermally driven VFf. Each demonstrator was actuated by a thermo-responsive SMA spring, which contracted and thus performed a closing motion when a critical temperature was reached. The tested demonstrators were placed in a climate chamber at three different temperatures. Videos were recorded from two perspectives (**A**,**D**), and the closing movement was further analyzed via motion tracking. Spring length was determined before and after the heating cycle (**D**), and the difference in length was calculated for complete closure. In (**D**), the upper orange line represents the demonstrator length and lower orange line the spring length before actuation. All boxplots above the red line indicate full closure (**B**). Each boxplot represents seven measurements. Different letters represent significant differences from other results with *p*-values below 0.05. Closing times (**C**) and speeds (**E**) were measured with the help of the motion paths (colored tracks of lobe tips). Closing times ranged from 36 s to 429 s with an average of 234.7 s, whereas average closing speeds ranged from 0.19 × 10^−3^ m/s up to 1.1 × 10^−3^ m/s. Each boxplot represents 14 measurements. In order to determine the force that was generated by the springs at closing temperature (65 °C), they were placed in the thermal chamber of the testing machine, and the output force was measured (**F**).

**Figure 7 biomimetics-07-00099-f007:**
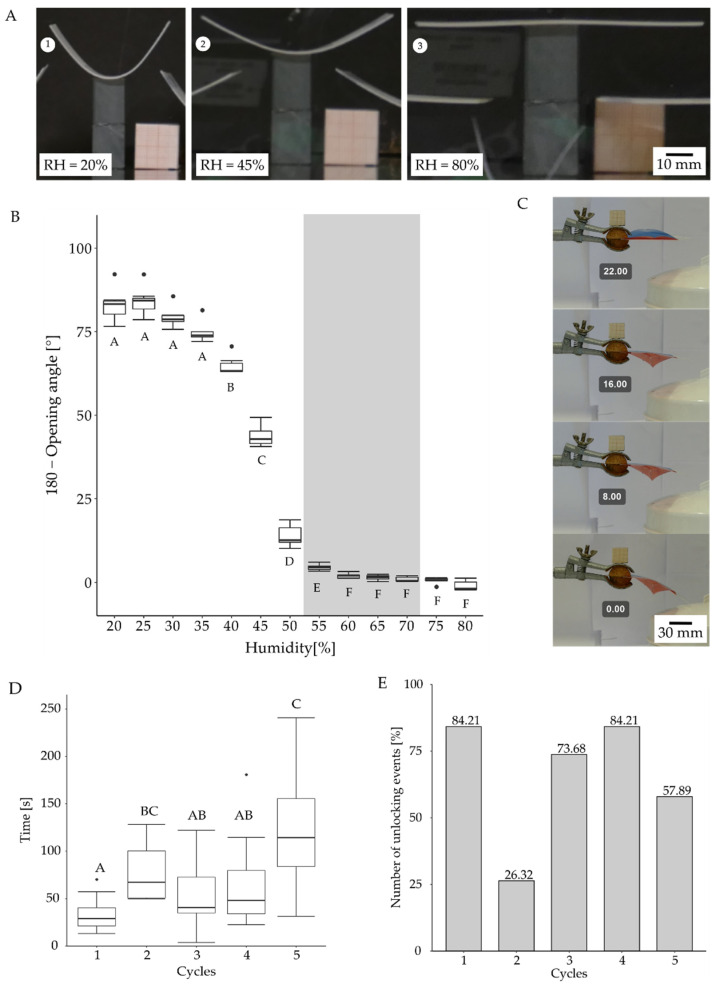
Autonomous environmentally triggered motion of VFf with a hydrogel-coated SMP backbone. In the dried state, the hydrogel-coated backbone is bent ((**A1**), 20%) but straightens when the level of humidity increases ((**A2**), 45%; (**A3**), 80%). At a constant temperature of 65 °C and 60% humidity, the backbone reaches 0° curvature (**B**) *n* = 7. The straightening process of the locked demonstrator (with foil attached) needs higher levels of humidity to overcome the resistance within the bent foil. Approximately 70% humidity is required to unlock the hydrogel demonstrator. To determine the unlocking time precisely from contact to switching, a stream of hot steam was directed selectively towards only the hydrogel-coated backbone (**C**). The time needed for the unlocking of the demonstrator ranged from 16.8 s to 241 s after first contact with the stream of hot water vapor (**C**,**D**). Significance levels are indicated by the capital letters next to each boxplot. Different letters represent significant differences from other results with *p*-values below 0.05. In this testing scenario, 19 demonstrators were tested five times. If the demonstrator did not unlock within 300 s, the test was declared unsuccessful (**E**).

**Figure 8 biomimetics-07-00099-f008:**
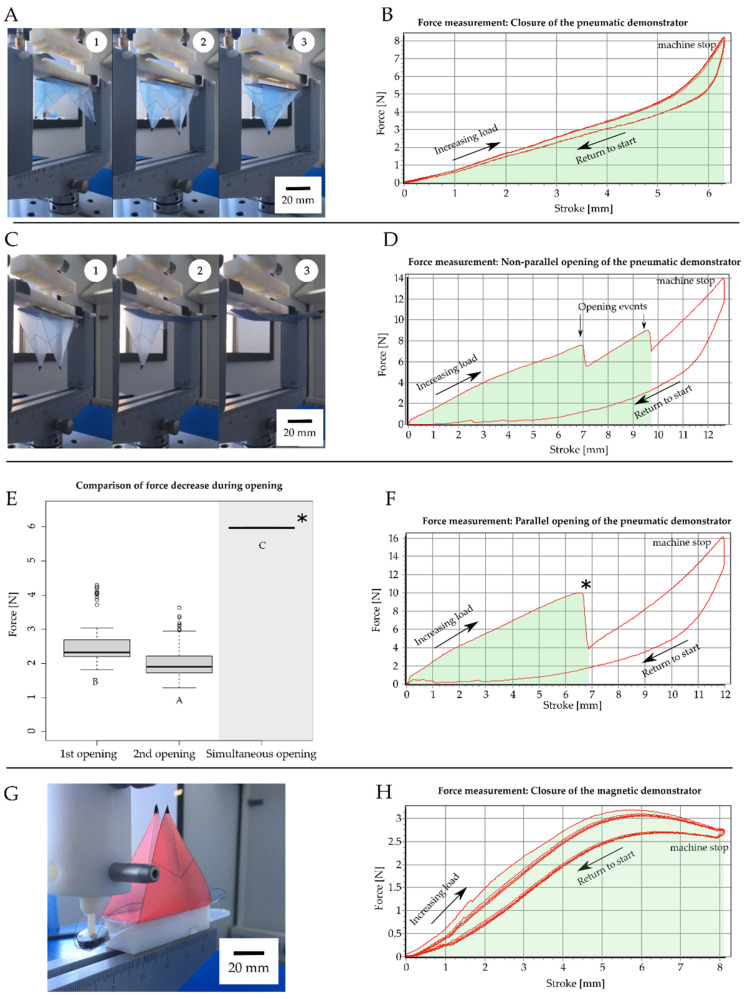
Kinematic energy measurements: examples of force displacement measurements carried out on the pneumatic and magnetic VFfs. Force measurements are used to calculate the energy needed to close and open the system. To imitate the closure of the pneumatic demonstrator, a piston with two pins is used to apply pressure to the backbone until the lobes close (**A1**–**A3**). The force (**B**) necessary to close the lobes is measured repeatedly. The force for the opening movement is measured with a piston designed to apply pressure at the center of the backbone (**C1**–**C3**). In most cases, the lobes open one after another (**C2**,**C3**) resulting in two distinct sudden drops in the force measured (**D**). When both lobes open simultaneously (**F**), the measured decrease in force (indicated by * in (**E**,**F**)) is larger compared with the separate opening events (**E**). Each of the three demonstrators used for video analysis were tested 25 times for opening and closing movements, respectively. The magnetic demonstrator (**G**,**H**) needed less force compared with the pneumatic demonstrator and also less energy. The force-displacement curves shown in (**B**,**D**–**F**,**H**) represent exemplary curves. Significance levels are indicated by the capital letters next to each boxplot. Different letters represent significant differences from other results with *p*-values below 0.05 (**B**).

**Table 1 biomimetics-07-00099-t001:** Comparison of *D. muscipula* with artificial Venus flytraps.

Schematic	Type	Actuation	Sensing	Snap-Buckling	Closing Time	Maximum Speed	Kinetic Energy Requirements for Actuation	Energy Consumption of the Test Setup	Efficiency of the Actuation	Reversibility
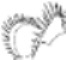	*Dionaea muscipula*	Stimulation of trigger hairs lead to active water displacement	Touch sensitive trigger hairs	Yes	0.15 s to 1.8 sLiterature: 0.1 s to 0.5 s [27]	0.016–0.245 m/s	Approx. 300 µmol ATP (at standard conditions equals 9.66 J) [27,28]	-	-	Yes
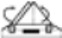	Pneumatic VFf	Pressurized air (approx. 0.7 bar)	No sensor/actuated manually	Yes	Closing:0.119 s to 0.311 sOpening: 0.023 s to 0.059 s	Opening movement: 3.26 m/s to 4.94 m/s	Opening: 38.49 mJ to 79.54 mJClosing: 19.04 mJ to 30.81 mJ	1 J for magnet valves and between 0.24 L and 0.3 L compressed air	3.7%	Yes
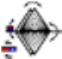	Magnetically driven VFf	Rotating magnetic field	No sensor/actuated manually	No	Lobes do not close completely	0.56 m/s to 3.56 m/s	11.52 to 17.15 mJ for manual closing	Between 27 and 39 J for the magnetic stirrer	5.6%	Yes
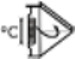	Thermally driven SMA VFf	Increase in temperature	Inherent to the material	No	36.0 s to 429 s with an average of 234.7 s.	0.000254 to 0.00117 m/s	Approx. 48.1 mJ to 315.9 mJ of energy provided by the SMA spring	5,464,800 J for thermal heating	3.3 × 10^−8^%	Yes
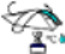	Hydrogel coated VFf	Change in humidity	Inherent to the material	No	No real closure but unlocking time: 33.2 s to 126 s	Not determined	Environmental humidity of approx. 77%	129,712.68 J for steam production		-

## Data Availability

Data is available in the supplementary material document and on request.

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
