# Peer review of "Novel Motion Sequences in Plant-Inspired Robotics: Combining Inspirations from Snap-Trapping in Two Plant Species into an Artificial Venus Flytrap Demonstrator"

_biomimetics, 2022, doi:10.3390/biomimetics7030099_

Round 1

Reviewer 1 Report

  It is a good work about plant inspired robot. The idea is novel and the realization process is excellent. It can inspire more work in this field. 

Author Response

Comments and Suggestions for Authors

It is a good work about plant inspired robot. The idea is novel and the realization process is excellent. It can inspire more work in this field. 

We thank reviewer 1 for his/her opinion and for this favorable comment.

Reviewer 2 Report

The study focuses on the novel motion sequences in plant-inspired robotics: combining inspirations from snap-trapping in two plant species into an artificial Venus flytrap demonstrator. The paper shows good theoretical and practical background.

My main attention to the paper is that:

1. The paper needs additional formatting in some places.

2. On the end of Equations must be "dot" or "comma".

3. Which type of programming language (software, Matlab, Python or other) was used in investigation? Please explain why?

4. The Table 1 is to big. 

5. Please explain more deeply this sentence (pg. 24/26., lines: 688-689 ) <A uniform resonance-like motion is achieved via actuation through rotating magnetic fields.>

Author Response

Comments and Suggestions for Authors

The study focuses on the novel motion sequences in plant-inspired robotics: combining inspirations from snap-trapping in two plant species into an artificial Venus flytrap demonstrator. The paper shows good theoretical and practical background.

My main attention to the paper is that:

  1. The paper needs additional formatting in some places.

We thank reviewer 2 for his/her comment, we checked the formatting in changed it in appropriate area, e.g. we added a tab to the figure 1 caption, so that the caption is not split between pages (p.4 li.106). We did the same for figure 4. The final copyediting will be performed by the journal type setters.

  1. On the end of Equations must be "dot" or "comma".

We added a dot at the end of equation (1).

  1. Which type of programming language (software, Matlab, Python or other) was used in investigation? Please explain why?

We thank reviewer 2 for his/her comment, in chapter 2.6 the statistical analysis is explained for which we used the statistic programming language R and software RStudio. We changed the sentence to “The software RStudio (v. 1.2.5042, R Core Team 2017) and statistic programming language R were used for statistical analysis and calculations.”(p.8 li. 276-277). For the movement analysis via video and picture-based analysis we used the Kinovea software as we described in chapter 2.1 (p. 4 li. 129-131). For the experiments, we used the specific software of the testing machines. We used these programs, as they are open-source and freely available.

  1. The Table 1 is to big. 

We thank reviewer 2 for his/her comment, as mentioned above the final editing will be done by journal and there are no regulations concerning format of tables. We have converted the table to horizontal format, but for this we had to rotate the table by 90°.

  1. Please explain more deeply this sentence (pg. 24/26., lines: 688-689 ) <A uniform resonance-like motion is achieved via actuation through rotating magnetic fields.>

Following the comment of reviewer 2, we explained and discussed this in more detail in chapter 4.3 of the discussion.

Reviewer 3 Report

The authors present a novel motion sequence for plant-inspired robotics that combines, for the first time, the motion principles of two snap-trap mechanisms, as observed in plants, into an artificial Venus flytrap. They have done a lot of work to drive the compliant foil structure by using the pneumatic action, magnetic field, SMA spring and hydrogel. Through experiments, they have studied the kinematic characteristics of AVF and made some statistical analysis. But there are still some problems.

Main comments:

1.The introduction session should be improved. The snap-trap mechanisms are similar to bistable/multi-stable mechanisms. The author should include the related references, such as:

   Zhong, Yong, Ruxu Du, Peng Guo, and Haoyong Yu. "Investigation on a new approach for designing articulated soft robots with discrete variable stiffness." IEEE/ASME Transactions on Mechatronics 26, no. 6 (2021): 2998-3009.

J. R. Raney, N. Nadkarni, C. Daraio, D. M. Kochmann, J. A. Lewis, and K. Bertoldi, "Stable propagation of mechanical signals in soft media using stored elastic energy," Proceedings of the National Academy of Sciences, vol. 113, pp. 9722-9727, 2016.

2. In page 7, to investigate the possibility of reaching the natural frequency and gaining a resonance effect in the oscillating flapping motion of the system, the authors studied the tests at speeds of 400, 700, 800, 900, 1000, and 1300 rpm, respectively, and I wondered why these speeds were chosen that way。

2. In page 14, section3.4.1, The authors tested five SMA springs to determine the critical temperature. The five springs begin to contract at different temperatures. What causes this to happen, are the parameters of these five springs not the same, or is it due to the difference in the location in the climate chamber? In addition, in P22, section4.4,The authors have considered that the VFF closest to the chamber ceiling first reaches the critical transition temperature. However, three SMA driven VFFS are still put in the same climate chamber for experiments. It doesn't seem very rigorous。

3.In page 18, section2.5, Is the typography here wrong? The previous section was 3.4.2. In addition, in this section the authors calculate the total energy consumed by the different drive modes andthe amount of kinetic energy (work) required either to open or to close the VFfs 。 So why not calculate the efficiency of each drive and compare it with the biological model. Efficiency is an important indicator in the field of bionic robots.

4. In page 22, section4.3, The authors claim that motion becomes more regular as the system approaches its assumed natural frequency of 800-900 rpm. There is a lack of theoretical analysis of natural frequencies, and only subjective judgments are made based on aesthetic and visual regularity.

5. The author tested the performance of VFF under four different driving modes and compared it with the biological model. However, it seems that the whole VFF system only tried different drives and showed advantages in some aspects, and did not form a perfect and comprehensive system.

Minor comments:

1.In page 20, line 548. There are two question marks in brackets. What does this mean, or is there a typographical error.

Author Response

1.The introduction session should be improved. The snap-trap mechanisms are similar to bistable/multi-stable mechanisms. The author should include the related references, such as:

   Zhong, Yong, Ruxu Du, Peng Guo, and Haoyong Yu. "Investigation on a new approach for designing articulated soft robots with discrete variable stiffness." IEEE/ASME Transactions on Mechatronics 26, no. 6 (2021): 2998-3009.

  1. R. Raney, N. Nadkarni, C. Daraio, D. M. Kochmann, J. A. Lewis, and K. Bertoldi, "Stable propagation of mechanical signals in soft media using stored elastic energy," Proceedings of the National Academy of Sciences, vol. 113, pp. 9722-9727, 2016.

Following the comment of reviewer 3, we have included the reference to the fact that the Venus flytrap can also be considered as a bistable system in the Introduction and have drawn the comparison to technical systems. There we have inserted the desired and additional references, see page 2 line 49 to 52.

  1. In page 7, to investigate the possibility of reaching the natural frequency and gaining a resonance effect in the oscillating flapping motion of the system, the authors studied the tests at speeds of 400, 700, 800, 900, 1000, and 1300 rpm, respectively, and I wondered why these speeds were chosen that way。

According to the comment of reviewer 3, we added the explanation to chapter 2.4 and stated the following on page 8 line 229 to 235:

“In preliminary tests, the rpm were steadily increased from 0 to 1500 and the behavior observed. From 400 rpm, the first clear movements were observed, at 800 and 900 rpm a uniform movement behavior was observed and from 1300 only irregular behavior. Because of this, the focus of the experiments was on the range of 800 and 900 rpm, and 100 rpm higher and lower were investigated. In addition, the extremes of 400 rpm (first clear movements) and 1300 rpm (only irregular irratic motions) were investigated.”

  1. In page 14, section3.4.1, The authors tested five SMA springs to determine the critical temperature. The five springs begin to contract at different temperatures. What causes this to happen, are the parameters of these five springs not the same, or is it due to the difference in the location in the climate chamber?

We thank reviewer 3 for his/her two-part comment. The differences in commercially available springs likely origin from the manufacturing process. The company assures consistent performance between their springs, but this does not seem to be the case here. During the measurement the five springs were all fixed at the same level in the climate chamber of the universal testing machine, so here is no thermal stratification effect at play.

We added a note to the differences in chapter 3.4.1 on page 15 line 399 to 400.

In addition, in P22, section4.4,The authors have considered that the VFF closest to the chamber ceiling first reaches the critical transition temperature. However, three SMA driven VFFS are still put in the same climate chamber for experiments. It doesn't seem very rigorous。

We thank reviewer 3 for his/her comment, we have placed three temperature sensors inside the chamber, one at the height of each demonstrator, so we could ensure to register the exact temperature at the location of each demonstrator. Thus, the thermal stratification and different times when the demonstrators started to move had no meaning for the actual investigation, since all demonstrators exact temperatures were recorded, and all showed the same behavior closing after passing the transition temperature.

However, we have included this note on purpose, since it belongs to our observations. We have added an exemplary measurement to the corresponding chapter of Supplementary Materials.

3.In page 18, section2.5, Is the typography here wrong? The previous section was 3.4.2. In addition, in this section the authors calculate the total energy consumed by the different drive modes and the amount of kinetic energy (work) required either to open or to close the VFfs . So why not calculate the efficiency of each drive and compare it with the biological model. Efficiency is an important indicator in the field of bionic robots.

We thank reviewer 3 for his/her comment, we are unsure what he/she means with the typography. If this comment is aimed at the numbering of the subchapters, this is specific to the main chapters and therefore does not run through all chapters. However, the headings always correspond to the same topic in terms of content.

An efficiency calculation for a biological is near to impossible, as it is highly dependent on the environmental conditions (temperature, energy input, etc.).We calculated the efficiency for the first three demonstrators system as we here have corresponding force measurements and input energy values. We added these values to table 1 and added the calculations to the supplementary materials document.

  1. In page 22, section4.3, The authors claim that motion becomes more regular as the system approaches its assumed natural frequency of 800-900 rpm. There is a lack of theoretical analysis of natural frequencies, and only subjective judgments are made based on aesthetic and visual regularity.

We agree with the reviewer, but such an analysis was not the focus of our characterization. For better understanding we added a frequency measurement to the supplementary materials.

  1. The author tested the performance of VFF under four different driving modes and compared it with the biological model. However, it seems that the whole VFF system only tried different drives and showed advantages in some aspects, and did not form a perfect and comprehensive system.

We thank reviewer 3 for her/his comment. In this work, we focused on the characterization of the actuator system to identify their capabilities and requirements. The integration of all of them in one system is the current focus of our follow up project starting end of the year. In this project we will combine the SMP/Hydrogel backbone with the SMA and Magnet system, but the integration of the pneumatic drive in a whole system is more complex so a redesign of the pneumatic actuation system is necessary. So the integration of all actuators in one system will be shown in follow up papers.

Minor comments:

1.In page 20, line 548. There are two question marks in brackets. What does this mean, or is there a typographical error.

Thank you, we deleted this typographical error.

Round 2

Reviewer 3 Report

The authors have addressed my comments.